# Physical Properties and Structural Changes of Myofibrillar Protein Gels Prepared with Basil Seed Gum at Different Salt Levels and Application to Sausages

**DOI:** 10.3390/foods9060702

**Published:** 2020-06-01

**Authors:** Chang Hoon Lee, Koo Bok Chin

**Affiliations:** Department of Animal Science, Chonnam National University, Gwangju 61186, Korea; rhichanghoon@naver.com

**Keywords:** basil seed gum, salt level, myofibrillar protein, sausage

## Abstract

The objective of this study was to evaluate physical properties and structural changes of myofibrillar protein gels with basil seed gum (BSG) at different salt levels and develop the low-salt sausages with BSG. Myofibrillar protein (MP) gels were prepared with or without BSG at different salt concentrations (0.15, 0.30, and 0.45 M). Cooking yield (CY, %), gel strength (GS, gf), viscosity, sulfhydryl contents, protein surface hydrophobicity, scanning electron microscopy (SEM), and Fourier transform infrared spectroscopy (FTIR) of MP were measured. Pork sausages were manufactured with 1% BSG at both low-salt (1.0%) and regular-salt (1.5%) levels. pH, color, expressible moisture (EM, %), CY, textural profile analyses, FTIR, sulfhydryl group, and protein surface hydrophobicity (μg) were measured for analyzing the properties of sausages. The addition of 1% BSG to MP gels increased CY and shear stress. Among treatments with different salt concentrations, MP at 0.30 M salt level with 1% BSG had higher GS than that at 0.15 M salt level with BSG. In microstructure, swollen structures were shown in MP gels with BSG. Although CY of sausage at the low-salt concentration (1.0%) decreased, regardless of the BSG addition, hardness values of sausages with regular-salt level increased with the addition of 1% BSG was added. Protein surface hydrophobicity and sulfhydryl contents of sausages increased with the addition of 1% BSG, resulting in higher hardness and lower springiness than those without BSG. These results suggest that BSG could be used as a water-binding and gelling agent in processed meats.

## 1. Introduction

Salt is a major ingredient for the manufacture of processed meat products. It can lead to good texture and flavor, and even microbial growth is inhibited. However, excessive usage of dietary salt can cause hypertension and induce the cardiovascular disease. The WHO [1] recommends that the consumption of sodium from food should be reduced to less than 2 g/day (i.e., sodium chloride of less than 5 g/day) for adults. Salt reduction in meat products causes detrimental effects on some properties such as water-holding capacity, texture, flavor, and shelf life [2]. The released moisture from meat products by salt reduction could be recovered for the properties of products by the addition of hydrocolloids.

Basil seed gum (BSG) is extracted from swelled basil seed by water. It contains a large amount of unesterified galacturonic acid [3]. BSG is composed of 43% glucomannan, 24% xylan, and 2% glucan, induced to improve the thickening and gelling properties [4]. It also forms thermos-irreversible gels mixed with protein [5]. Many researchers had studied physical properties of BSG in food systems. Hosseini-Parvar et al. [6] reported that processed cheese products with BSG showed high firmness and low melting ability. Afshar Nik et al. [7] found that the mixed structure between BSG and protein matrices decreased the syneresis of yogurt and increased the firmness. BSG applied to meat products could decrease the cooking loss of products during the heating due to the extremely high water-binding capacity from BSG. Even if the low-salt meat products are manufactured, BSG could bind a lot of water molecules in meat protein matrices during the heating and the detrimental effects on the properties of meat products were not observed. Therefore, the objective of this study was to evaluate physicochemical properties and structural changes of myofibrillar protein gels, and sausages with BSG at different levels of salt.

## 2. Materials and Methods

### 2.1. Materials

Pork loin and ham (Landrace x Yorkshire, grade A, Korea) were bought from a local meat market (Samho Co., Gwangju, Korea). Unnecessary parts for the processing, such as visible fat and connective tissue, were removed by trimming process. The trimmed pork loin was prepared by forming 1–2 cm^3^ cubes, and then stored in a −50 °C freezer until usage. Pork hams were ground using a meat grinder (M-12s, Fuji Plant, Busan, Korea) with a 32 mm grind plate. Basil seeds were soaked in double-distilled (dd) water for 30 min and the ratio of water to seed was 100:1. After completely swelling, the gum layer was taken from the seed surface by the extractor (NNJ-1415JM, Hurom, Gimhae, Korea). Basil seed gum (BSG) in the wet state was freeze-dried at −50 °C for 48 h. Dried BSG samples were ground using a model HMF-3260S mixer (Hanil Electric, Seoul, Korea) to obtain a uniform particle size.

### 2.2. Preparation of Myofibrillar Protein Gels

Pork loin cubes were ground with a buffer solution at 0.1 M NaCl and 50 mM phosphate using a food grinder (HR-2160, Phillips, Seoul, Korea) to extract the salt soluble protein. The precipitate was obtained after centrifuged at 3000 rpm for 15 min. These procedures were repeated three times. The emulsion was mixed with 0.1 M NaCl buffer solution and filtered using cheesecloth to obtain salt soluble proteins. The filtered solution was centrifuged at 3000 rpm for 15 min again. The final precipitate emulsion was mixed with a buffer solution to uniformly 4% protein concentrate solutions were prepared. The protein concentrate solutions were mixed with or without BSG as following the formulation (Table 1). Each mixed sample was put into vial tubes and those were heated from 20 to 80 °C using a water bath (WB-22, Daihan Scientific Co., Seoul, Korea). After completed the heating procedure, samples were placed into an ice water for 30 min and stored at 4 °C until analyzed.

### 2.3. Manufacture of Sausages

Sausages were manufactured with 1.0% BSG powder at two different salt concentrations (1.0 and 1.5%) as shown in Table 2. Ground pork ham and curing agents were mixed and ground by a food cutter (HMC-401, Hanil Electric, Seoul, Korea) for 3 min. The mixtures were stuffed into 45 mL plastic tubes heated to 72 °C monitored using stainless steel K type thermocouples inserted into the geometric center of the tube. Heated sausage samples were placed into ice water to completely cool down the temperature. Chilled sausage samples were stored at 4 °C in a refrigerator before usage.

### 2.4. Experimental Methods to Determine the Physical Properties of Myofibrillar Protein Gels

Cooking yield was measured based on the weight differences before and after heating. Gel strength was evaluated with a single cycle compression by an Instron Universal Testing Machine (Model#3344, Canton, MA, USA). The speed of the probe was 500 mm/min by steel drill chuck (33BA 1/2-20, Jacobs chuck, Sparks Glencoe, MD, USA). For measuring the viscosity of MP mixtures, mixture was loaded into the container of a rheometer probe. Shear stress was measured using a concentric cylinder type rotational rheometer (Model#RC30, Rheotec Messtechnik GmbH, Ottendorf-Okrilla, Germany). Shear rate was set at a range of 0 to 600/s.

### 2.5. Experimental Methods to Determine the Physical Properties of Sausages

pH and color values were measured on the internal surfaces of samples. The pH values of samples were measured at five different parts with a pH meter (Model#MP120, Mettler-Toledo, Schwarzenbach, Switzerland). Color values (CIE L*, a*, b*) of samples were measured at six different sections with a color reader (Model#CR-10, Minolta, Japan). Cooking yield (CY, %) was determined based on the weight differences between before and after cooking, as previously described in cooking conditions. After samples for expressible moisture (EM, %) were cut into cube shape of 1.5 g, and they were wrapped with three pieces of filter paper (Whatman #3) and put into the 45 mL plastic tubes. After centrifuging samples at 3000 rpm for 15 min, EM was calculated based on the weight differences between before and after centrifugation as described previously [8]. Textural properties were evaluated using by a Universal Testing Machine. Sausage samples were punctured 12.5 mm in height. Each sample was compressed twice to three-quarters of its height. Textural properties were measured 10 times on the randomly selected locations by a 500 N load cell at a cross speed of 300 mm/min. Results are represented as hardness (N), springiness (mm), gumminess, chewiness, and cohesiveness according to the method of Bourne [9].

### 2.6. Experimental Methods of Structural Changes in Myofibrillar Protein Gels or Sausages

Microstructures of samples were captured using a scanning electron microscopy (Model#JSM-6610LV microscope, JEOL Ltd., Tokyo, Japan). Samples were cut into cubes (3 mm^3^) and fixed with 2.5 % glutaraldehyde solution at 4 °C. Those fixed samples were placed into an osmium tetroxide solution for 5 h. After completely rinsing those samples, dehydration was performed from low to high concentration of ethanol. Dried samples were coated with gold by a coater (Model#108 autos putter coater, Cressington Scientific Instruments Ltd., Watford, England). In addition, the structure of samples was measured at 1000 magnification. The quantitative level of protein secondary structures was measured using Fourier transform infrared spectroscopy (Frontier FT-IR/NIR Spectrometer, PerkinElmer). Graphic data were prepared with wavelength ranged from 1750 to 1450 cm^−1^ to indicate secondary structure of protein matrix. Protein surface hydrophobicity from muscle protein was measured using a hydrophobic chromophore bromophenol blue (BPB) solution [10]. Unheated samples 1 g (20 mM phosphate buffer, control) and BPB 0.5 mL were mixed in 15 mL plastic tubes. After vortex those for 10 min, the mixtures were centrifuged at 3000 rpm for 15 min and got the supernatant to measure at 595 nm against a phosphate buffer (blank) on a spectrophotometer (UV-1601, Shimadzu, Kyoto, Japan). The index of hydrophobicity was obtained with the following formula: BPB bound (μg) = 500 μL × (OD of control − OD sample)/OD control. Sulfhydryl content was measured with the modified Ellman’s method using 5,5′-dithiobis-(2-nitrobenzoic acid) (DTNP) solution [11]. DTNP 0.5 mL, Tris 1 mL, and dd-water 8.4 mL were mixed with sample 0.1 g (20 mM phosphate buffer (pH 6.25), control). After incubating at room temperature for 5 min, mixtures were measured with absorbance value at 412 nm. Sodium dodecyl sulfate-polyacrylamide gel electrophoresis (SDS-PAGE) was performed to determine polymerization of sample. Ten percent of acrylamide separating gel and 4% stacking gel were used for loading samples [12]. Samples were diluted at 1% protein concentration, and mixed with sample buffer before loading into acrylamide gel. After loading the sample into acrylamide gels, proteins were separated at 150 V for 1.5 h. Pre-stained protein standard marker (161-0318, Bio-Rad, Hercules, CA, USA) was used to determine molecular weight of proteins in samples.

### 2.7. Statistical Analysis

Each experiment was performed in triplicate. Experimental data that evaluated MP were analyzed by two-way (2 levels of BSG × 3 levels of salt) analysis of variance (ANOVA), whereas data from sausages were analyzed by one-way ANOVA using SPSS 23.0 program (SPSS Inc., Chicago, IL, USA). Significant level was performed at *p* < 0.05.

## 3. Results and Discussion

### 3.1. Physical Properties of Myofibrillar Protein Gels as Affected by BSG and Salt Levels

Cooking yield (CY) and gel strength (GS) results of MP gels with or without BSG at different salt concentrations are shown in Table 3. The addition of BSG on MP gel increased the CY compared to MP gel without BSG. As the presence of hydrophilic groups in BSG, free water could be hold in meat matrix [13], resulting in increasing CY of MP gels with BSG. The CY of MP gel at 0.15 M salt level was lowest value as compared to those with higher salt levels. The GS of MP gel with BSG increased compared to that of MP gel without BSG. Although MP gels had no gelling ability at the salt level below 0.30 M, the combination of MP mixture and BSG resulted in well aggregation and formed a viscoelastic gel matrix. BSG polymer interacted and formed a three-dimensional structure, indicating thixotropic behavior of BSG [14]. Rafe et al. [15] reported that BSG did not have gelling ability under heating conditions. However, BSG could form the gel when cooling down at 20 °C. The amount of unsubstituted mannan regions from BSG can increase junction zones of the gel during heating or freezing conditions, resulting in increased hardness of gel [16]. Different ionic strength from additional levels of salt also affected GS and CY of MP mixture with BSG.

Figure 1 shows viscosity of MP mixtures with or without BSG at different salt concentrations. The addition of BSG to MP mixture increased shear stress by increasing shear rate. In particular, MP mixture with BSG at 0.30 M salt level significantly increased shear stress value. MP mixture with increased salt levels showed high shear stress compared to lower amount of salt in MP mixture. Terrell [17] also reported that the addition of salt affected the hydration of protein, resulting in improved water-holding capacity and viscosity of meat batter. Hosseini-Parvar et al. [18] reported that the addition of salt increased BSG emulsions, resulting in aggregation. BSG emulsion at low ionic strength can decrease the viscosity and viscoelasticity of BSG emulsion, as compared to that at high ionic strength [19]. Amide and Mirhoseeini [20] reported that the combination of protein and polysaccharide by a covalent linkage could increase viscosity. Although the combination of protein and BSG increased shear stress, no synergetic effects with the addition of salt and BSG were observed.

### 3.2. Structural Changes in Myofibrillar Protein Gels as Affected by BSG and Salt Levels

Protein surface hydrophobicity and sulfhydryl contents of MP mixtures with or without BSG at different salt concentrations are shown in Table 4. MP mixtures with BSG increased hydrophobic surface of samples. Hydroxyl groups from carbohydrate prefer to interact with two molecules of water. When the levels of protein surface hydrophobicity are increased, it caused to higher water-holding capacity. MP mixtures at 0.45 M salt level had the higher protein surface hydrophobicity than those at lower salt levels (< 0.30 M). As shown in Table 4, MP mixtures with increasing salt concentration showed reducing the sulfhydryl (SH) contents. Metal ion from sodium chloride has a screening effect on the negative charge of protein surface. As a result, high salt levels can lead to strong formation of disulfide bonds [21]. However, MP mixtures with BSG showed the increase of sulfhydryl content. Hosseini-Parvar et al. [18] reported that adsorption ability and structural stability are decreased when ionic strength is decreased by strong hydrophobicity of BSG. 

Figure 2 shows microstructures of MP gels with or without BSG. MP gels with BSG at low-salt concentration (<0.3 M salt) showed swollen structure, indicating increased water-binding ability and gelling property (Figure 2c,d). MP gels at 0.45 M salt level showed compact and dense structure compared to those at lower salt levels, regardless of addition of BSG. The structure of scattered cotton was found in the microstructure of BSG, which has fibril and globular structure for absorbing free water [15]. Hosseini-Parvar et al. [6] reported that the BSG gel network acts as tension agent on protein matrix, indicating that BSG enhance gelling properties and viscosity of MP in the present study. 

BSG increases the mechanical properties of MP gels with correlation between meat protein and BSG. With increasing salt concentrations, the band intensities at the peaks of 1650 cm^−1^, 1624 cm^−1^, and 1680 cm^−1^ (α-helix/ unordered structures and β-sheet) decreased based on quantitative analysis results as shown in Figure 3. These peaks’ intensities were also decreased in MP with BSG, due to its well-formed structure. Figure 4 shows the area of 3000–2800 cm^−1^ of MP gels with or without BSG and the peak area was affected by different levels of salt. This area indicates the stretching state of C-H bonds of methyl groups [5], and its intermolecular interaction between BSG and MP was related to each other. MP gels with BSG had lower peak intensity than those without BSG. The addition of BSG at 0.15 M salt level not only had higher peak intensity than those at higher salt levels, but also had higher peak intensity than MP gels without BSG at higher salt levels. These results suggested that salt level at the above 0.30 M affected intermolecular interaction between BSG and MP in gels. 

Figure 5 shows protein patterns of MP mixtures with or without BSG at different salt concentrations. The intensity of myosin heavy chain (MHC) in MP mixture with BSG decreased compared to those without BSG. At the band above 195 KDa protein, MP mixtures without BSG showed a band regardless of salt level. Specific protein band also appeared between 67 and 97 KDa due to incorporate BSG to MP mixture. Bombrun et al. [22] reported that the contents of myosin were not influenced by increasing salt level. However, the contents of actin decreased with increasing salt levels. The ratio of actin to myosin decreased with increasing salt level, resulting in high cohesiveness of gel.

pH and color values of sausages are shown in Table 5. Regardless of salt levels, there were no differences in pH and color values of samples between sausage with and without BSG. Ramírez et al. [23] reported that general kinds of hydrocolloids did not affect the color. The expressible moisture (EM, %) of sausages did not differ among treatments, as shown in Table 6. While the sausages with BSG at 1.5% salt level showed the highest CY, sausages without BSG at 1.0% salt level had the lowest CY (*p* < 0.05). CYs of sausages without BSG at 1.5% salt level were similar to those of sausages with BSG at 1.0% salt level (*p* > 0.05). Textural properties of sausage are presented in Table 7. Sausages without BSG at 1.5% salt showed the lowest hardness value among all treatments. Comparing sausages with salt level at 1.5% salt with and without BSG, the addition of BSG to sausage showed higher hardness value as compared to those without BSG. However, no difference in hardness value was observed between sausages with and without BSG at 1.0% salt level. Zhang et al. [24] found that the increase in ion strength enhanced hydrogen bonds between protein and water molecules. However, hydrogen bonds among proteins are reduced, resulting in weaker MP gels. Sausages containing BSG decreased springiness values which showed the lowest value at 1.5% salt level among all treatments.

### 3.3. Structural Changes of Sausages

Protein surface hydrophobicity (PSH) and sulfhydryl (SH) contents of sausages are shown in Table 8. PSH of sausages with BSG at 1.5% salt had the highest value among all sausages (*p* < 0.05). Sausage without BSG at 1.0% salt level had the lowest value of PSH (*p* < 0.05). MP gels with high ionic strength can be induced to expose hydrophobic residues from interior protein molecules. Thus, hydrophobicity of protein increased by increasing salt concentration [25]. Kaewmanee et al. [26] also reported that PSH was proportional to salt concentration. There was no significant difference in SH content among different salt concentrations (*p* > 0.05). At the same salt concentration, the addition of BSG into sausage increased SH content, resulting in lower amounts of disulfide bonds. These results indicated that both factors, such as salt concentration and BSG, influenced PSH and SH contents, resulting in higher hardness and lower springiness of sausage with BSG than sausage without BSG. Bombrun et al. [22] reported that increasing salt concentration caused to reduced free thiol contents and high adhesiveness between pork meat pieces. Figure 6 shows quantity of secondary protein in sausage s with or without BSG at different salt levels. Quantitative analysis of sausages without BSG revealed that intensities of bands at 1650 cm^−1^, 1624 cm^−1^, and 1680 cm^−1^ (α-helix/unordered structures and β-sheet) decreased with increasing salt concentration. However, the band intensities of sausages with BSG were not different among different salt concentrations.

## 4. Conclusions

MP gels with BSG at 0.30 M salt level had higher CY and GS than those without BSG at 0.45 M salt level. Sausages with BSG at 1.0 % salt level increased physical properties, as compared to those without BSG at 1.5% salt level. Based on these results, BSG could be used as a water-binding agent in low-salt meat products without having detrimental effects on their physicochemical properties.

## Figures and Tables

**Figure 1 foods-09-00702-f001:**
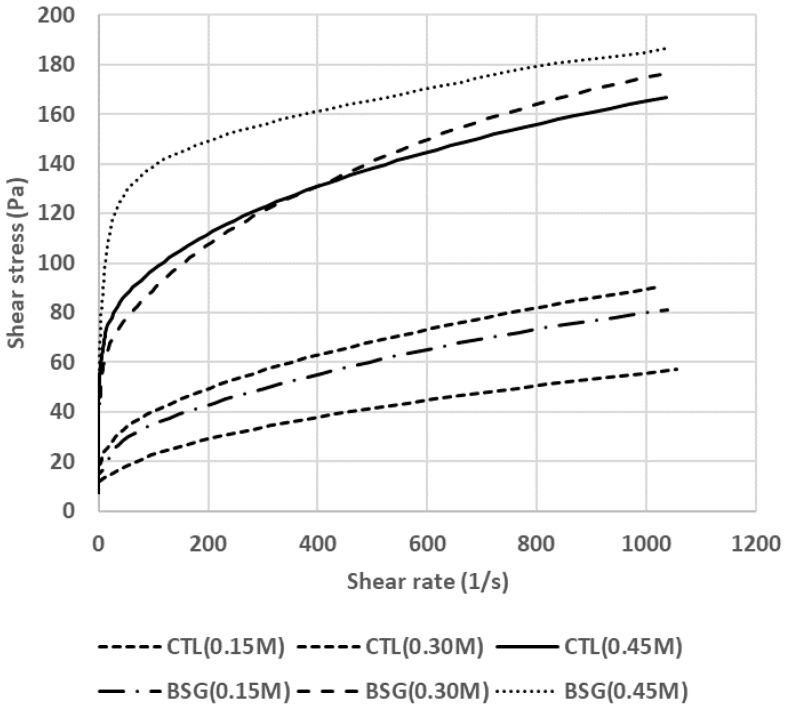
Viscosity of myofibrillar protein mixtures with basil seed gum at different salt concentrations. Treatments: CTL(0.15 M), myofibrillar protein mixtures (MP) without basil seed gum (BSG) at 0.15 M salt concentration; CTL(0.30 M), MP without BSG at 0.30 M salt concentration; CTL(0.45 M), MP without BSG at 0.45 M salt concentration; BSG(0.15 M), MP with BSG at 0.15 M salt concentration; BSG(0.30 M), MP with BSG at 0.30 M salt concentration; BSG(0.45 M), MP with BSG at 0.45 M salt concentration.

**Figure 2 foods-09-00702-f002:**
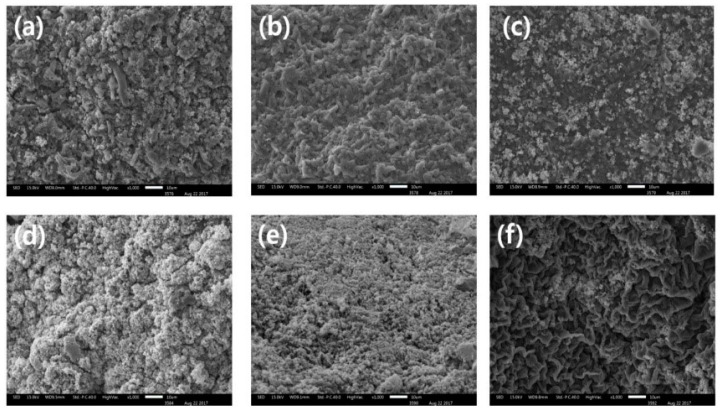
Microstructure of myofibrillar protein with basil seed gum at different salt concentrations (**a**) Control (0.15 M), (**b**) Control (0.30 M), (**c**) Control (0.45 M), (**d**) BSG (0.15 M), (**e**) BSG (0.30 M), (**f**) BSG (0.45 M). **Treatments:** CTL(0.15 M), myofibrillar protein mixtures (MP) without basil seed gum (BSG) at 0.15 M salt concentration; CTL(0.30 M), MP without BSG at 0.30 M salt concentration; CTL(0.45 M), MP without BSG at 0.45 M salt concentration; BSG(0.15 M), MP with BSG at 0.15 M salt concentration; BSG(0.30 M), MP with BSG at 0.30 M salt concentration; BSG(0.45 M), MP with BSG at 0.45 M salt concentration.

**Figure 3 foods-09-00702-f003:**
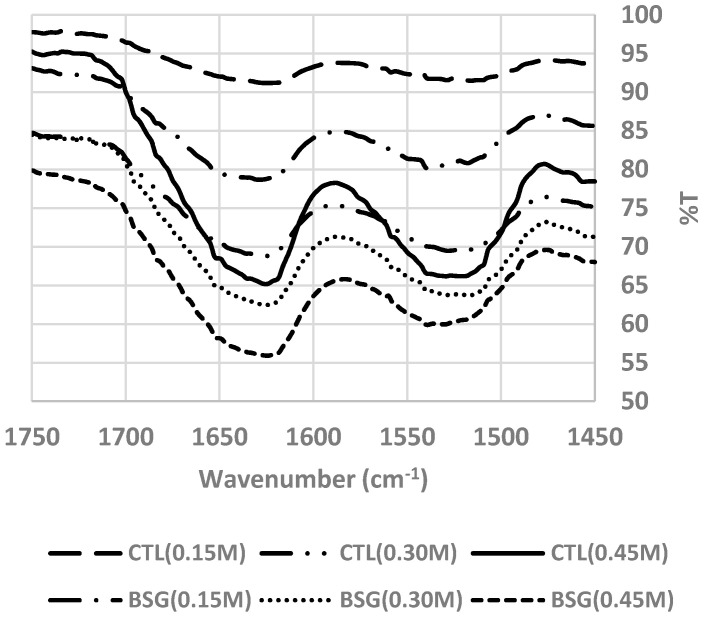
FTIR (wavenumber 1750–1450 cm^−1^) of myofibrillar protein with basil seed gum at different salt concentrations. **Treatments:** CTL(0.15 M), myofibrillar protein mixtures (MP) without basil seed gum (BSG) at 0.15 M salt concentration; CTL(0.30 M), MP without BSG at 0.30 M salt concentration; CTL(0.45 M), MP without BSG at 0.45 M salt concentration; BSG(0.15 M), MP with BSG at 0.15 M salt concentration; BSG(0.30 M), MP with BSG at 0.30 M salt concentration; BSG(0.45 M), MP with BSG at 0.45 M salt concentration.

**Figure 4 foods-09-00702-f004:**
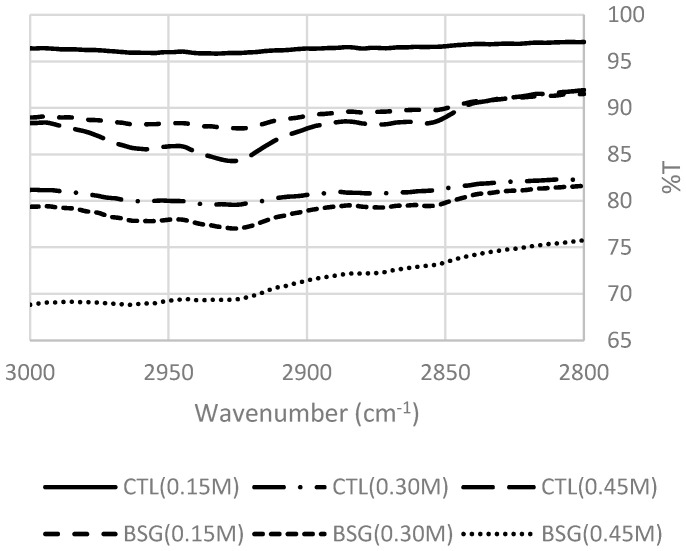
FTIR (wavenumber 3000–2800 cm^−1^) of low-fat sausages with basil seed gum at different salt levels. **Treatments:** CTL(0.15 M), myofibrillar protein mixtures (MP) without basil seed gum (BSG) at 0.15 M salt concentration; CTL(0.30 M), MP without BSG at 0.30 M salt concentration; CTL(0.45 M), MP without BSG at 0.45 M salt concentration; BSG(0.15 M), MP with BSG at 0.15 M salt concentration; BSG(0.30 M), MP with BSG at 0.30 M salt concentration; BSG(0.45 M), MP with BSG at 0.45 M salt concentration.

**Figure 5 foods-09-00702-f005:**
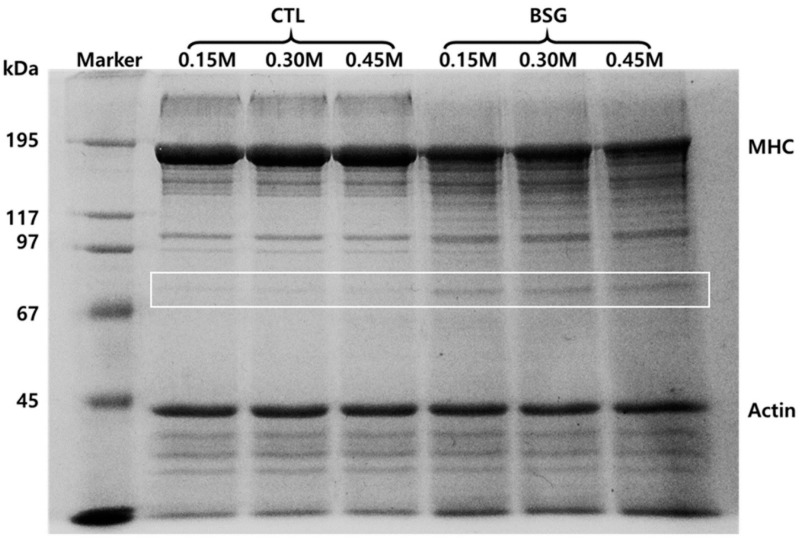
SDS-PAGE of myofibrillar protein with basil seed gum at different salt concentrations. **Treatments:** CTL(0.15 M), myofibrillar protein mixtures (MP) without basil seed gum (BSG) at 0.15 M salt concentration; CTL(0.30 M), MP without BSG at 0.30 M salt concentration; CTL(0.45 M), MP without BSG at 0.45 M salt concentration; BSG(0.15 M), MP with BSG at 0.15 M salt concentration; BSG(0.30 M), MP with BSG at 0.30 M salt concentration; BSG(0.45 M), MP with BSG at 0.45 M salt concentration.3.3. Physical and Textural Properties of Sausages

**Figure 6 foods-09-00702-f006:**
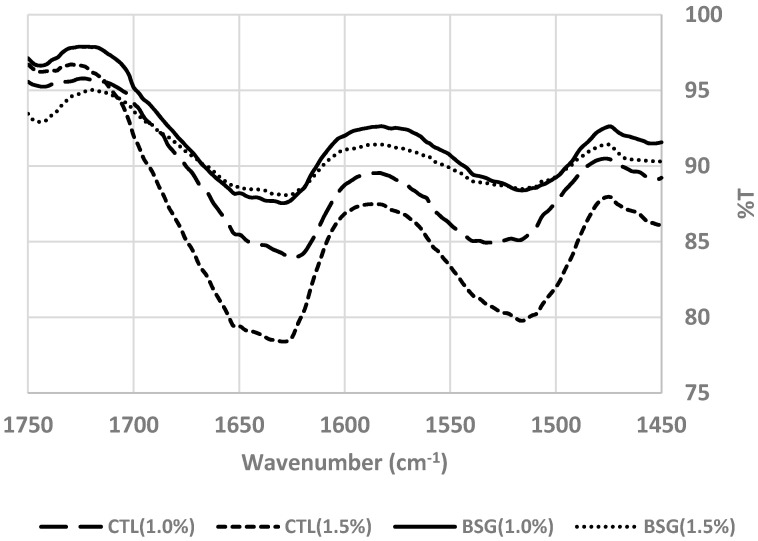
FTIR (wavenumber 1750–1450 cm^−1^) of low-fat sausages with basil seed gum at different salt levels. Treatments: CTL(1.0%), sausage without basil seed gum (BSG) at 1.0% salt concentration; CTL(1.5%), sausage without BSG at 1.5% salt concentration; BSG(1.0%), sausage with BSG at 1.0% salt concentration; BSG(1.5%), sausage with BSG at 1.5% salt concentration.

**Table 1 foods-09-00702-t001:** Formulation of myofibrillar protein gels with basil seed gum at different salt levels.

Ingredients (mg/mL)	Control			Basil Seed Gum		
	0.15 M	0.30 M	0.45 M	0.15 M	0.30 M	0.45 M
Myofibrillar protein	40.0	40.0	40.0	40.0	40.0	40.0
Buffer solution	10.0	10.0	10.0	9.50	9.50	9.50
Basil seed gum	0.00	0.00	0.00	0.50	0.50	0.50
Total	50.0	50.0	50.0	50.0	50.0	50.0

**Table 2 foods-09-00702-t002:** Formulation for the manufacture of low-fat sausages with basil seed gum at different salt levels.

Treatments (Salt Level, %)	Control		Basil Seed Gum	
	1.0	1.5	1.0	1.5
1. Meat	60.0	60.0	60.0	60.0
2. Water	38.5	38.5	38.5	38.5
3. Salt	0.80	1.30	0.80	1.30
4. Sodium tripolyphosphate	0.40	0.40	0.40	0.40
5. Cure blend *	0.25	0.25	0.25	0.25
6. Sodium erythorbate	0.05	0.05	0.05	0.05
7. Basil seed gum	0.00	0.00	0.50	0.50
Total	100.0	100.5	100.5	101.5

* Cure blend, 93.75% salt + 6.25% sodium nitrite (NaNO_2_).

**Table 3 foods-09-00702-t003:** Cooking yield and gel strength of myofibrillar protein mixtures with basil seed gum at different salt concentrations.

		Salt Concentration		
		0.15 M	0.30 M	0.45 M
Cooking yield (%)	CTL *	67.3 ± 4.06 ^B,b^	84.9 ± 0.03 ^B,a^	86.7 ± 3.41^B,a^
	BSG *	98.7 ± 0.38 ^A,a^	100 ± 0.05 ^A,a^	100 ± 0.00 ^A,a^
Gel strength (gf)	CTL	15.6 ± 0.76 ^B,b^	15.9 ± 3.89 ^B,b^	109 ± 3.73 ^B,a^
	BSG	91.0 ± 2.14 ^A,b^	122 ± 3.95 ^A,a^	122 ± 10.8 ^A,a^

^a,b^ Means (*n* = 3) with the same superscripts in a same row are not different (*p* > 0.05). ^A,B^ Means (*n* = 3) with the same superscripts in a same column are not different (*p* > 0.05). * CTL, control; BSG, basil seed gum.

**Table 4 foods-09-00702-t004:** Protein surface hydrophobicity and sulfhydryl (SH) contents of myofibrillar protein mixtures with basil seed gum at different salt concentrations.

	Treatments *		Salt Concentrations		
	CTL	BSG	0.15 M	0.30 M	0.45 M
Protein surface hydrophobicity	34.6 ± 2.48 ^B^	45.1 ± 2.72 ^A^	18.9 ± 6.09 ^b^	28.5 ± 8.10 ^b^	72.1 ± 1.10 ^a^
SH content	33.1 ± 2.38 ^B^	45.3 ± 1.95 ^A^	65.2 ± 5.01 ^a^	33.7 ± 1.18 ^b^	18.8 ± 7.71 ^c^

^a–c^ Means (*n* = 3) with the same superscripts in salt concentrations are not different (*p* > 0.05). ^A,B^ Means (*n* = 3) with the same superscripts in treatments are not different (*p* > 0.05). * Treatments: CTL, control; BSG, basil seed gum.

**Table 5 foods-09-00702-t005:** pH and color values of low-fat sausages with basil seed gum at different salt levels.

Treatments (Salt Levels, %)	Control		Basil Seed Gum	
	1.0	1.5	1.0	1.5
pH value	5.97 ± 0.01	5.96 ± 0.03	5.99 ± 0.01	5.95 ± 0.02
CIE L *	74.6 ± 0.95	75.1 ± 1.28	74.7 ± 1.14	74.1 ± 0.31
CIE a *	9.86 ± 0.60	9.07 ± 0.19	9.69 ± 0.55	9.48 ± 0.09
CIE b *	4.42 ± 0.19	4.22 ± 0.14	5.06 ± 0.37	4.89 ± 0.29

* CIE L, Lightness; * CIE a, Redness; * CIE b, Yellowness

**Table 6 foods-09-00702-t006:** Expressible moisture and cooking yield of low-fat sausages with basil seed gum at different salt levels.

Treatments (Salt Levels, %)	Control		Basil Seed Gum	
	1.0	1.5	1.0	1.5
Expressible moisture (%)	38.4 ± 0.43 ^a^	38.4 ± 3.07 ^a^	37.0 ± 1.38 ^a^	35.1 ± 1.48 ^a^
Cooking yield (%)	86.8 ± 1.13 ^c^	93.0 ± 1.03 ^b^	93.1 ± 0.53 ^b^	97.3 ± 0.14 ^a^

^a–c^ Means (*n* = 3) with same superscripts in a same row are not different (*p* > 0.05).

**Table 7 foods-09-00702-t007:** Textural properties of low-fat sausages with basil seed gum at different salt levels.

Treatments (Salt Levels, %)	Control		Basil Seed Gum	
	1.0	1.5	1.0	1.5
Hardness (gf)	2955 ± 71.1 ^a^	2606 ± 159 ^b^	3066 ± 75.6 ^a^	2924 ± 43.9 ^a^
Springiness (mm)	6.89 ± 0.42 ^a^	7.26 ± 0.13 ^a^	6.09 ± 0.13 ^b^	5.16 ± 0.25 ^c^
Gumminess	30.8 ± 8.95 ^a^	22.7 ± 2.58 ^a^	22.7 ± 0.72 ^a^	23.2 ± 0.83 ^a^
Chewiness	195 ± 34.5 ^a^	164 ± 21.5 ^a^	142 ± 4.15 ^a^	116 ± 7.52 ^a^
Cohesiveness	8.33 ± 0.12 ^a^	8.52 ± 0.06 ^a^	8.03 ± 0.43 ^a^	8.17 ± 0.04 ^a^

^a–c^ Means (*n* = 3) with same superscripts in a same row are not different (*p* > 0.05).

**Table 8 foods-09-00702-t008:** Protein surface hydrophobicity and SH contents of low-fat sausages with basil seed gum at different salt levels.

Treatments (Salt Levels, %)	Control		Basil Seed Gum	
	1.0	1.5	1.0	1.5
Protein surface hydrophobicity	12.5 ± 0.52 ^d^	13.6 ± 0.00 ^c^	16.3 ± 0.24 ^b^	18.5 ± 0.52 ^a^
SH content	26.3 ± 1.72 ^b,c^	22.8 ± 1.77 ^c^	31.3 ± 0.68 ^a^	27.7 ± 0.65 ^a,b^

^a–d^ Means (*n* = 3) with same superscripts in a same row are not different (*p* > 0.05).

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
