# Peer review of "Physical Properties and Structural Changes of Myofibrillar Protein Gels Prepared with Basil Seed Gum at Different Salt Levels and Application to Sausages"

_foods, 2020, doi:10.3390/foods9060702_

Round 1
Reviewer 1 Report
The Authors have investigated an interesting topic and the theme has been properly described. I would like to congratulate Authors for the good-quality of the article, the literature reported used to write the paper, and for the clear and appropriate structure. The manuscript is well written, presented and discussed, and understandable to a specialist readership. In general, the organization and the structure of the article are satisfactory and in agreement with the journal instructions for authors. The subject is adequate with the overall journal scope.
The work shows a conscientious study in which a very exhaustive discussion of the literature available has been carried out. The introduction provides sufficient background, and the other sections include results clearly presented and analyzed exhaustively.
However, I suggest to revise Tables and Figures by adding as footnote the description of the used acronyms.
Author Response
The Authors have investigated an interesting topic and the theme has been properly described. I would like to congratulate Authors for the good-quality of the article, the literature reported used to write the paper, and for the clear and appropriate structure. The manuscript is well written, presented and discussed, and understandable to a specialist readership. In general, the organization and the structure of the article are satisfactory and in agreement with the journal instructions for authors. The subject is adequate with the overall journal scope.
--> Thank you
The work shows a conscientious study in which a very exhaustive discussion of the literature available has been carried out. The introduction provides sufficient background, and the other sections include results clearly presented and analyzed exhaustively.
--> Thank you
However, I suggest to revise Tables and Figures by adding as footnote the description of the used acronyms.
-->Thank you and revised according to the comments
Reviewer 2 Report
Good experimental design and the research issue is important to the meat industry. The authors have been very thorough and the research methods are excellent. They have done a very good job of presenting the results and discussing the results. There are some minor English issues and a few details that need to be included as defined below.
The abstract provides an overview of the experiment and the results.
Line 50 Reword to remove the contraction.
Line 59 Reword ie. Pork ham meat was ground using a 32 mm grinder plate (M-12s, ….
Line 63 Replace ‘at the wet’ with ‘in the wet’
Line 64 Replace ‘for having a uniform particle size’ with ‘to obtain a uniform particle size’
Line 68 Weren’t these pork loin cubes? Was water used to carry the NaCL and phosphate or was it added dry?
Line 70-72 Reword ie., The emulsion was mixed with 0.1 M NaCl buffer solution and filtered using cheesecloth to obtain salt soluble proteins.
Do not start a sentence with ‘And’. Reword please.
Line 73 Replace ‘prepare 4% concentrations of protein’ with ‘4% protein concentrate solutions were prepared’
Line 74 Reword ‘Then, it was mixed with’ with ‘The protein concentrate solutions were mixed with’
Table 1 is very good, please remove the line in the header between control and basil seed gum at columns 4 and 5 so that the line goes over the respective treatments only.
Table 2. please see comment for Table 1. Please use indentation in table to differentiate levels of non-meat ingredients to make it clear that these are a subset of the 3.
Line 83 Replace ‘and those were heated by time to reach 72°C of the geometric center of sample’ to ‘heated to 72°C monitored using DEFINE TYPE thermocouples inserted into the geometric center of the tube.’
Line 85 Define the dimensions of the plastic tubes.
Line 94 Insert ‘the’ before probe and define the type and size of probe and the load cell used. Was this a single cycle compression? Assume so, but would be nice to define.
Line 105 What were the dimensions of the cubes?
Lines 126 to 130 Please reword this sentence.
Line 136 Please do not start a sentence with a number, but spell the number out.
Lines 143 to 146. Were least squares means calculated? What mean separation test was used. For the two-way analysis, was the interaction term included in the model?
Please check English and sentence structure for Results section.
I would like to see p-values in tables and were only mean separations are shown – Tables 3, 4,8, some measure of error – either SD, SE or RMSE.
Author Response
Reviewer 1 (blue color)
Good experimental design and the research issue is important to the meat industry. The authors have been very thorough and the research methods are excellent. They have done a very good job of presenting the results and discussing the results. There are some minor English issues and a few details that need to be included as defined below.
The abstract provides an overview of the experiment and the results.
Line 50 Reword to remove the contraction.-->Changed (Line 50)
Line 59 Reword ie. Pork ham meat was ground using a 32 mm grinder plate (M-12s, -->Changed (Line 59-60)
Line 63 Replace ‘at the wet’ with ‘in the wet’ --> Changed (Line 63)
Line 64 Replace ‘for having a uniform particle size’ with ‘to obtain a uniform particle size --> Changed (Line 64)
Line 68 Weren’t these pork loin cubes? --> Changed (Line 67) Was water used to carry the NaCL and phosphate or was it added dry? --> Changed (Line 67-68)
Line 70-72 Reword ie., The emulsion was mixed with 0.1 M NaCl buffer solution and filtered using cheesecloth to obtain salt soluble proteins.
Do not start a sentence with ‘And’. Reword please. --> Changed (Line 69-71)
Line 73 Replace ‘prepare 4% concentrations of protein’ with ‘4% protein concentrate solutions were prepared’--> Changed (Line 72-73)
Line 74 Reword ‘Then, it was mixed with’ with ‘The protein concentrate solutions were mixed with’ --> Changed (Line 73)
Table 1 is very good, please remove the line in the header between control and basil seed gum at columns 4 and 5 so that the line goes over the respective treatments only. --> Changed (Table1)
Table 2. please see comment for Table 1. Please use indentation in table to differentiate levels of non-meat ingredients to make it clear that these are a subset of the 3. ---> Changed (Table2)
Line 83 Replace ‘and those were heated by time to reach 72°C of the geometric center of sample’ to ‘heated to 72°C monitored using DEFINE TYPE thermocouples inserted into the geometric center of the tube.’ -->Changed (Line 85-86)
Line 85 Define the dimensions of the plastic tubes. --> Changed (Line 84)
Line 94 Insert ‘the’ before probe and define the type and size of probe and the load cell used. Was this a single cycle compression? Assume so, but would be nice to define.--> Changed (Line 91-94)
Line 105 What were the dimensions of the cubes? _prepared the cubes based on the weight.
Lines 126 to 130 Please reword this sentence. --> Changed (Line 126-130)
Line 136 Please do not start a sentence with a number, but spell the number out. -->Changed (Line 136)
Lines 143 to 146. Were least squares means calculated? What mean separation test was used. For the two-way analysis, was the interaction term included in the model?
--> Yes, the interaction term was included in the model. If they had interaction, it was separated by treatment within BSG or BSG within a treatment and they were analyzed by Duncan'rs multiple reange test with three replications.
Please check English and sentence structure for Results section.
--> It was checkeed the English grammer
I would like to see p-values in tables and were only mean separations are shown – Tables 3, 4,8, some measure of error – either SD, SE or RMSE. --> Changed (All tables)
Thnak you for the suggestions!
Shoud you have further questions, please le us know.
